# Mouse Mammary Tumor Virus (MMTV) and MMTV-like Viruses: An In-depth Look at a Controversial Issue

**DOI:** 10.3390/v14050977

**Published:** 2022-05-06

**Authors:** Francesca Parisi, Giulia Freer, Chiara Maria Mazzanti, Mauro Pistello, Alessandro Poli

**Affiliations:** 1Dipartimento di Scienze Veterinarie, Università di Pisa, Viale Delle Piagge, 2, 56124 Pisa, Italy; francesca.parisi@vet.unipi.it; 2Dipartimento di Ricerca Traslazionale e delle Nuove Tecnologie in Medicina e Chirurgia, Università di Pisa, Via Savi 10, 56126 Pisa, Italy; giulia.freer@unipi.it (G.F.); mauro.pistello@unipi.it (M.P.); 3Fondazione Pisana per la Scienza, Via Ferruccio Giovannini, 13, 56017 San Giuliano Terme, Italy; c.mazzanti@fpscience.it

**Keywords:** lymphoma, mammary carcinoma, mouse mammary tumor virus, p14, PCR, superantigen

## Abstract

Since its discovery as a milk factor, mouse mammary tumor virus (MMTV) has been shown to cause mammary carcinoma and lymphoma in mice. MMTV infection depends upon a viral superantigen (sag)-induced immune response and exploits the immune system to establish infection in mammary epithelial cells when they actively divide. Simultaneously, it avoids immune responses, causing tumors through insertional mutagenesis and clonal expansion. Early studies identified antigens and sequences belonging to a virus homologous to MMTV in human samples. Several pieces of evidence fulfill a criterion for a possible causal role for the MMTV-like virus in human breast cancer (BC), though the controversy about whether this virus was linked to BC has raged for over 40 years in the literature. In this review, the most important issues related to MMTV, from its discovery to the present days, are retraced to fully explore such a controversial issue. Furthermore, the hypothesis of an MMTV-like virus raised the question of a potential zoonotic mouse–man transmission. Several studies investigate the role of an MMTV-like virus in companion animals, suggesting their possible role as mediators. Finally, the possibility of an MMTV-like virus as a cause of human BC opens a new era for prevention and therapy.

## 1. History and Identification of MMTV

In 1936, John Bittner discovered a carcinogenic “milk factor” that could be transmitted through milk from a mouse, bearing mammary tumors to offspring that, once adult, developed mammary tumors [1]. Viral particles were first identified as cancer agents in mouse milk in 1949, when Graff et al. demonstrated that they could cause mammary tumors if injected intraperitoneally into laboratory mice [2]. In 1966, the virus was identified as a retrovirus, and thanks to Duesberg and Blair it became known as mouse mammary tumor virus (MMTV) [3]. The president of the USA, Richard Nixon, allocated considerable financial support to study this causative agent; therefore, MMTV has been intensively studied and proven to be a causal agent of mammary cancer in wild and experimental mice. An important step forward was the identification of an RNA homologue of MMTV RNA in human BC [4]. MMTV has been intensively studied and has been proven to be a causal agent of mammary cancers in wild and experimental mice. It was subsequently identified in various murine organs and tissues, including the prostate, the salivary glands, and the lymphatic system [5]. It was hypothesized that human endogenous retrovirus (HERV) sequences, being very similar to MMTV, may be the remains of the virus integrated into the human genome over the millennia. Subsequently, it was shown that the differences between the MMTV *env* gene and HERV were sufficient to differentiate them by PCR [6,7]. In 2001, the identification of an entire proviral sequence from two human BC patients in the USA, bearing 95% homology to MMTV, but only 57% to HERVs [8], supported the idea that an exogenous virus (called human mammary tumor virus, HMTV) similar to MMTV was present in some tumor samples. Over time, the presence of viral sequences from a virus homologous to MMTV was confirmed in humans and highlighted also in dogs, cats, and monkeys [6,7,9,10,11,12,13,14,15].

## 2. Classification and Viral Properties

MMTV belongs to the Retroviridae family, subgenus Betaretrovirus, and it is the prototype of slow-transforming retroviruses. Virions of MMTV exhibit a “B-type” morphology with prominent surface spikes and an eccentric condensed core. Its genome is 8–10 kb in size and, like all retroviruses, is characterized by long terminal repeats (LTRs) at the 3′ and 5′ ends in its proviral form. MMTV LTRs are exceptionally long (approximately 1.3 kb) because they encode two additional genes: *sag*, encoding a viral accessory protein that functions as a superantigen located at the 3′ end of the genome, overlapping U3, and *rem*, which encodes an RNA export protein, translated from a doubly spliced mRNA overlapping the *env* gene. The *sag* gene is absent from other members of the genus, but several other viruses encode activities analogous to REM [16]. LTRs are potent transcriptional enhancers. At least one region located near the 5′ end of LTRs is called the mammary gland enhancer [17,18]. In addition to this region, the MMTV LTRs contain other regulatory elements, including multiple hormone response elements (HREs), believed to be involved in upregulation of virus production [8,19], and several negative regulatory elements (NREs), preventing transcription from the LTR in tissues where MMTV is not expressed [20]. Before the differentiation of the mammary gland, MMTV LTRs are repressed by the transcriptional factor CCAAT displacement protein (CDP) [21], which is subsequently down-regulated upon mammary gland differentiation [22]. It is likely that CDP represses promoters by displacing positive regulatory factors and associating with histone deacetylases. MMTV LTRs also contain transcription enhancer factor-1 (TEF-1) binding sites, one canonical high activity site near the CDP binding site, and other less relevant activity sites elsewhere [20]. TEFs, also known as TEAD transcription factors, are known to play a promoting role in several cancer types because they regulate multiple genes strongly correlated with tumorigenesis [23].

Due to the encoding of non-structural proteins, MMTV, which was originally classified as a simple retrovirus, is now classified as a complex retrovirus [24,25]. There are at least five transcripts generated by the MMTV genome. First of all, a full-length unspliced RNA is packaged into the virions as a viral genome. This transcript also encodes for Gag, Pol, and pro-dUTPase (DUT) protein. The function of dUTPase (dUT) in viral replication is still not clear and may have to do with its replication in non-dividing cells. MMTV Env protein is encoded by a single spliced mRNA in mice, which produces a signal peptide (SP p14), surface (SU gp52), and transmembrane domain (TM gp36). The SU protein is generated by removal of the signal peptide by signal peptidase and cleavage of the transmembrane domain by cellular Furin. There are at least two other alternatively spliced mRNAs encoding for Sag and Rem. Sag plays an essential role in MMTV-associated tumor genesis [26] and is encoded by two different transcripts. Rem regulates export of MMTV RNAs from the nucleus; it is required for the efficient transport of unspliced viral RNA by associating with a Rem-responsive element located on MMTV RNA [27].

## 3. Life Cycle

Viral particles are shed in mothers’ milk and transmitted to the suckling pups through the intestinal epithelium. Here, the virus penetrates M cells in the epithelial layer and invades the underlying lymphoid tissue [28]. The virus first infects dendritic cells and B lymphocytes in Peyer’s patches of the gut [27,29,30]. B lymphocytes and dendritic cells process viral Sag protein, a type II transmembrane glycoprotein, and then express it in association with major histocompatibility complex (MHC) class II on the cellular surface, presenting antigens to CD4+ T cells. Sag-activated T cells proliferate and produce cytokines, stimulating and recruiting additional dendritic, B, and T cells. In this way, the expansion of the different cell clones takes place, and the result is the proliferation of both infection-competent and infected cells. Once they reach the mammary gland, they infect cells [27].

Therefore, both B and T lymphocytes play critical roles in MMTV infection [30]. The long latency between the ingestion of infected milk and mammary tumor development suggests that MMTV does not contain an oncogene. For this reason, it is presently classified as a non-acute transforming retrovirus [27,30,31]. In blood, MMTV activity can be detected as a subviral form as intracytoplasmic particles associated with nucleated blood cells, while B particles are not detected [30,32]. The mechanism of tumorigenesis proposed starts with the infection of a single mammary stem cell. Mammary stem cells have the potential to form two epithelial lineages in the mammary gland: myoepithelial cells, found outside of the ducts, and ductal and alveolar luminal cells. MMTV-induced tumors develop from the second ones, giving rise to a pregnancy-dependent hyperplastic alveolar nodule (HAN) at first, followed by hormone-independent mammary tumorigenesis [27,33]. It is known that MMTV also infects a variety of other epithelial tissues, including the salivary gland, kidney, seminal vesicle, epididymis, and testis [34,35,36,37]. It is likely that infection occurs in the same way as in the mammary epithelium (Figure 1), but only the latter is transformed subsequently to infection and replication of MMTV with few exceptions [38]. 

The regenerative activity of the mammary epithelium is deemed to play a role because it increases roughly 30 times at each pregnancy [39,40]; as a consequence, multiple pregnancy cycles lead to quicker mammary tumorigenesis [41,42,43]. This, per se, does not completely explain the occurrence of neoplastic transformation in the mammary epithelium because almost all MMTV-infected cells die by apoptosis at involution. However, it is likely that the persistence of MMTV-infected multipotent mammary stem cells, which are self-renewing, can acquire additional mutations and give rise to MMTV-induced hyperplasia and tumors [40].

MMTV exists as two forms in mice. The one discussed above is known as the exogenous virus, acquired by susceptible strains through milk. Differently, other strains inherit endogenous copies of the provirus [32]. Virtually all laboratory strains have from 2 to 8 endogenous proviruses (termed Mtv loci and numbered in order of their discovery) [44]. In general, endogenous proviruses do not lead to the production of functional viruses, although a few inbred strains have retained active endogenous proviruses; an example may be the GR strain, selected for its high mammary tumor incidence [45]. These active endogenous proviruses might represent recent germline integrations, whereas it has been estimated that other endogenous Mtvs have been present in the mouse genome for 20 million years [46,47].

MMTV infects mammary epithelial cells when they are driven to divide, both during puberty and pregnancy [27]. After binding to Transferrin receptor 1 (TfR1) on the cellular surface, it is internalized into a low-pH endosome. After un-coating, the viral RNA is reverse-transcribed, transported to the nucleus, and integrated as proviral DNA into the genome of the host cell [27]. Viral mRNAs are translated on cellular ribosomes. The primary translation products of retroviral *gag* and *env* mRNAs are polyproteins that are subsequently cleaved to give rise to mature Gag and Env proteins [48]. Translation products, together with progeny RNA, are assembled at the cell membrane into viral particles (the so-called B particles) that are released from the cell by budding from the plasma membrane. Finally, maturation takes place by proteolytic cleavage of virion polyproteins by viral and cellular proteases [49].

## 4. Mechanisms of Mammary Tumor Oncogenesis

Because MMTV does not carry oncogenes, it cannot produce rapidly appearing, polyclonal tumors upon expression [49,50]. Tumor induction occurs through the activation of sets of cellular oncogenes driven by promoters or enhancers present on viral LTRs [27,51]. MMTV-induced tumor growth follows two steps: it starts from a pre-neoplastic hormone-dependent lesion, known as the pregnancy-dependent HAN, and progresses to behaving as a hormone-independent tumor [5,27,43,52]. The hypothesis that mammary tumors and hyperplasia develop from stem cells and, therefore, represent the progeny of mutated stem cell populations, is supported by the identification of tumors with single proviral insertions [5,53,54]. At present, a wealth of evidence supports the concept that MMTV-induced premalignant and malignant lesions develop from infection of a single stem cell that acquires transforming mutations and expands clonally after infection [40,55,56,57,58,59,60]. Several different MMTV-specific common integration sites (CIS) have been implicated in mammary tumorigenesis. Genes whose expression has been frequently altered because of MMTV integration include members of *wnt* (-1/10b; -3, -3a), *fgf* (-3, -4, -8), *rspo* (-2, -3), *notch4/int3*, and *eIF3e/int 6* (*Int 6*) [5,27,60,61]. Particularly, Callahan and Smith (2008) [61] referred to *wnt*, *fgf*, and *rspo* as “core” CIS, due to their high frequency of occurrence. The *wnt* signaling pathway is the main target of tumor cell outgrowth [55]. It is involved in the determination of cell and tissue polarity, stimulation of cell proliferation and differentiation, and tissue homeostasis [61]. Fgfs are 28 multifunctional peptides acting as growth factors [5,60,62], while Rspo and Notch family proteins are involved in signaling pathways for cellular proliferation [63,64]. Rspo synergize with Wnt in the stabilization of β-catenin in the traditional Wnt signal pathway. The Notch gene family encode transmembrane proteins involved in cell fate decision during development. Otherwise, MMTV insertions in introns of the eIF3e gene in the opposite transcriptional orientation lead to truncated RNA (Int6sh) that causes the persistence of alveolar hyperplasia and increases the predisposition to mammary tumorigenesis in vivo [61]. In principle, MMTV integration events may occur upstream, 5′ of the target gene, in the opposite transcriptional orientation, or downstream, 3′ of the target gene, in the same transcriptional orientation [54,65]. Viral integrations events occur infrequently within gene coding regions, but often at a significant distance from the target gene, up to more than 200 kb away [66,67,68]. Even if further studies are needed, several studies have suggested that MMTV-induced mammary tumors are the result of different multiple integrations [27].

## 5. Mechanisms of Oncogenesis—Lymphoid Tumors

An MMTV variant known as type B leukemogenic virus (TBLV) was reported to cause T cell lymphomas in susceptible mice [69,70]. If compared with MMTV, TBLV differs in the structure of the LTR. The U3 region of TBLV LTR has a deletion of 440 bp, resulting in the loss of negative regulatory elements (NREs) and a duplication or triplication of flanking sequences. This is a common feature not only of TBLV, but also of most MMTV variants isolated from T cell lymphomas [71,72,73,74,75,76]. Indeed, the deletion of NREs and the multiplication of regulatory elements within LTRs are characteristic of retroviral enhancers [77,78,79,80], but, in the case of TBLV, they encode for T cell-type specific transcriptional enhancers [81]. These cell-type specific enhancers, together with the deletion of NREs, are sufficient to change MMTV pathogenicity from mammary tumors to T-cell lymphomas [82]. Furthermore, the insertion model proposed for MMTV is also typical of TBLV [54,65,83,84]. Dudley et al., (2016) identified at least four CIS for TBLV in the *myc*, *rorc*, *notch1*, and *tblvi1* gene regions, but most TBLV integrations were found upstream or downstream *rorc and myc* [5]. Even if studies on the subject are very few, TBLV-induced T-cell lymphomas were hypothesized to be polyclonal [5,82], based on the observation that most of the TBLV insertions near *c-myc* could not be detected by Southern blotting, but only through a PCR strategy. This suggested that these tumors were made up of a heterogeneous population of cells, each with an independent TBLV insertion near the oncogene. Moreover, a number of tumors showed multiple different integration events, indicating the polyclonal nature of these neoplasms. A possible explanation for this may be seen in the fact that the virus variant encoded a truncated Sag protein that may act as a transcriptional element causing polyclonal proliferation of T cells. These, in turn, may allow additional TBLV infection events, providing additional proviral integrations [82].

## 6. Susceptibility and Resistance

The mechanism of MMTV infection is complex and the outcome is highly dependent on the host’s immune response, so that breakpoints that modify host susceptibility can occur at any step of the immune response itself. Several host genes are known to be implicated in susceptibility or resistance to MMTV infection in mice, but the individual and intrinsic aspects that mostly affect the outcome of infection are linked to MHC class II molecules, the Vβ domain of TCR and APOBEC3 proteins, with different mechanisms. First of all, when milk-borne viral particles enter the small intestine, B lymphocytes and dendritic cells are the initial targets of viral infection [85]. As already discussed, infected antigen-presenting cells (APCs) express virus-encoded Sag through MHC class II molecules [86]. Different MHC class II molecules show different presentation capabilities. I-E molecules are known as the best presenters for all the described MMTV Sags [51]. For this reason, due to poor Sag presentation, mouse strains lacking MHC class II I-E genes are relatively resistant to infection by most MMTVs [87]. Secondly, in contrast to other antigens recognized by both the Vβ and the Vα domains of TCR when presented by MHC class II, Sag recognition is predominantly achieved through a number of defined Vβ domains, resulting in the stimulation of approximately 10% to 30% of T cells. Further interactions between activated T cells and B cells induce the expression of costimulatory molecules on B cells, which promotes further activation and proliferation of T cells and B cells [88].

Previous studies report that MMTV exploits the innate immune Toll-like receptor 4 (TRL4) in order to induce self-tolerance, achieved using a well-characterized ligand to trigger TRL4, the lipopolysaccharide (LPS). Binding of MMTV to LPS results in the production of the immunosuppressive cytokine interleukin 10 (IL10), required to quench the antiviral response. This is an example of how microbiota may promote viral propagation and transmission, thus resulting in an innovative evasion strategy used by the virus [89,90]. 

Differently from the exogenous virus, endogenous proviruses are not tumorigenic (except for Mtv-1 and Mtv-2) [91], but, rather, may protect against further viral infection. *Mtv*-encoded Sag stimulates the deletion of Sag-specific Vβ+ T cell subsets during the formation of the immune repertoire in utero [5]. This occurs presumably through a mechanism of self-tolerance, because endogenous Sag acts like a self-antigen and causes the deletion of cells that might be self-reactive [92]. Infection by exogenous virus carrying a Sag of the same Vβ specificity as the endogenously integrated proviruses cannot take place due to lack of responding T cells and consequent resistance to the infection. A certain degree of resistance is also typical of mice with mutations in TLR4, such as C3H/HeN mice, which show a decrease in milk-borne transmission and a delay in mammary tumor development, if compared with animals without mutations in the same genes [5,93,94]. 

MMTV infection can also be restricted by the intrinsic immune factor apolipoprotein B editing complex 3 (APOBEC3). APOBEC3 proteins are cytidine deaminase enzymes that convert cytidine into uridine in the single-stranded DNA retroviral replication intermediate [95], thereby damaging DNA prior to genomic integration [85]. To act on DNA, APOBEC3 proteins should be captured and packaged into retroviral particles before their release from infected cells, interfering with the replication of the retrovirus once it enters a new target cell [96]. In addition, APOBEC3 proteins efface MMTV infection at two levels: their expression in lymphoid cells avoids the spread to the infection within the host, while their expression in myoepithelial cells decreases the infectious power of the viral particles released into milk. Another acquired mechanism of defense, typical of I/LnJ mice, is the production of neutralizing antibodies coating viral particles, thus making infectious milk-borne MMTV avirulent and preventing the infection of nursing pups [97]. However, animals with deficiencies in the pattern recognition receptors involved in TLR7 and/or the adaptor/s thereof, as well as I/LnJ mice, carrying an immune response allele in the recessive gene negatively affecting antibody production, cannot produce neutralizing antibodies [5,98,99,100]. 

It should be noted that not all mechanisms of resistance to MMTV depend on the immune response. A certain degree of susceptibility can also be linked to mutations induced by MMTV insertion and the signaling pathways where the target genes participate, which can be more involved than others in the progression from pre-neoplastic to malignant lesion [101]. It was also shown that NH mice were resistant to the virus, possibly due to hormonal changes [102].

## 7. MMTV Transmission to Human Species: The Human Mammary Tumor Virus

Since the discovery of MMTV, several scientists have hypothesized for a similar virus in the etiology of human BC [27]. BC is the most common neoplasm among women worldwide [103]. It affects more than two million women, accounting for around half a million deaths annually [104]. Even if estrogen and radiation exposure are now considered important predisposing factors, BC is a complex and heterogeneous disease, and its precise causes are still unclear [103]. A large body of scientific evidence accumulated over several decades led to the hypothesis that a virus was etiologically involved in sporadically-occurring BC. However, the results from different research groups were contradictory, and its role in this disease remained elusive for years [6,10]. 

The earliest evidence for the involvement of a virus similar to MMTV in human BC came from the electron microscopy identification of viral particles resembling those of MMTV in milk samples from healthy women. Later, these particles were shown to have reverse transcriptase activity. Following these observations, MMTV-like sequences were identified in human BC samples by hybridization analyses, and immunologic assays brought to light MMTV proteins and antibodies against MMTV in biopsies or sera from BC patients. Furthermore, serum from patients bearing BC was capable of partially inhibiting infection by MMTV in mice. In the 1990s, molecular techniques allowed the detection of MMTV *env*-like sequences (MMTVels) in a high percentage of human breast carcinomas [105]. The putatively responsible virus became known as HMTV—human mammary tumor virus [8]. Viral proteins were detected in human BC cells containing MMTVels. A complete provirus was found and partially sequenced. Hybridization with PCR and in situ PCR were subsequently used in further studies, allowing specific amplification of the target MMTV-like gene. In several studies, sequencing confirmed specific amplification of sequences with high homology to MMTV. Interestingly, these sequences showed genetic variability, testifying to the absence of murine contamination.

In 2005, Indik et al. showed, first, that MMTV could infect human cells [24], and then that human cells could support replication of MMTV [106], thus suggesting that human cells are permissive for MMTV. These were two essential prerequisites to hypothesize that an infectious agent similar to MMTV might play some role in human BC. A study in 2010 reported that, in humans, MMTV-like virus could be detected in between 14% and 74% of breast cancers [107]. However, over time, the results of these studies were questioned. Indeed, other groups worldwide could find no evidence of this virus in BC tissue nor any association between HMTV and human BC. Others regarded positive detection of HMTV in samples as environmental contamination, or as non-specific [6,10]. As a matter of fact, the search for an MMTV-like virus has been complicated by the potential presence of HERVs integrated in the human genome, as well as *Mtv* in the murine one, that required differentiating from possible MMTVels. Several HERVs have accumulated mutations over the years, that made them non-coding, while, on the other hand, some others have retained the ability to produce viral products [108,109,110]. Accordingly, HERVs had been grouped into three classes based on the extent and the type of homology to retroviruses: Class I, grouping gammal- and epsilon-like retroviruses; Class II, comprising betaretrovirus-like retroviruses; and Class III, with spuma-like retroviruses [111]. The endogenous retroviruses belonging to Class II with sequences similar to exogenous MMTV are grouped in the HERV-K supergroup, in turn consisting of ten groups, named with the acronym HML (human MMTV-like) followed by a number from 1 to 10 [112]. Sequences from viruses belonging to HML groups could lead to false positive results when the presence of exogenous MMTV is investigated, due their high similarity. The recent studies by Lessi et al. have demonstrated that an MMTV-like betaretrovirus has been present in humans since the Copper Age. These studies took specific precautions to rule out the possibility of contamination and of false-positives due to endogenous-retrovirus integration into the human genome [14]. 

As discussed above, it is generally accepted that MMTV is involved in murine mammary tumorigenesis by viral DNA insertions in the vicinity of genes controlling cell growth. Despite various cellular genes known to be affected by MMTV proviral insertions in mice that are deregulated or mutated in primary mammary tumors [67], no evidence of MMTV/HMTV proviral insertions in the vicinity of these genes were found in DNA from human BC tissue, nor did next generation sequencing succeed in detecting viral sequences in BC samples [113]. Given this, mechanisms other than insertions for HMTV tumorigenicity, already documented for other retroviruses, were proposed. First of all, it was suggested that MMTV could encode an oncogenic protein equivalent to the Tax protein or to the bZIP factor (HBZ) encoded by Human T-Lymphotropic Virus-1 (HTLV-1). The former activates a cascade of events leading to CD4+ T lymphocytes survival and proliferation and to interference with cell senescence, thus leading to CD4+ T lymphocyte transformation [114], while the latter supports the growth of human T-cell lines [115]. However, to date, MMTV does not appear to encode an oncogenic protein, equivalent to Tax or HBZ, able to directly transform mammary epithelial cells.

Another point of investigation is the potential additional functions of envelope proteins, in comparison to certain other retroviruses encoding for envelope proteins directly involved in oncogenic transformation [31]. On this basis, MMTV *env* overexpression, together with an immunoreceptor tyrosine-based activation motif (ITAM) originally described by Katz et al., 2005, has been implicated in the transformation of mammary epithelial cells grown in three-dimensional culture [116] and in mice [117]. Further proof of a certain degree of involvement of ITAM in tumorigenesis came from the observations that an infectious MMTV provirus lacking a functional ITAM developed mammary tumors at a higher frequency and with longer latency [5]. A recent study suggests that the ITAM domain acts by suppression of cellular apoptosis thorough ITAM-mediated Src tyrosine kinase signaling [118]. Moreover, a cleavage product of MMTV envelope precursor protein known as p14 was also suggested to be involved in tumorigenesis, depending on its phosphorylation status (see below) [119]. Other authors suggest that already known proteins, such as Rem, Sag, the negative active factor (Naf) [118], or TEADs transcriptional factors [20] could have additional functions linked to transformation. Another conceivable hypothesis may be that MMTV infections activate latent human DNA viruses involved in BC or may facilitate infection by a second virus suspected to be involved in BC, such as the Epstein–Barr Virus or human papillomavirus [120,121,122]. Finally, the interaction between MMTV/HMTV and human endogenous retroviruses may also play a role in neoplastic transformation of breast epithelial cells [123]. Whether or not a putative MMTV-like virus is linked to human BC through one or more of these mechanisms has yet to be verified. 

Even if compelling evidence demonstrated that an MMTV-like virus is linked to human BC, the origin of this virus is still controversial. Szabo et al. [10] hypothesized that humans could acquire MMTV through contact with mice or cats, based on the evidence that an MMTV-like virus could also infect felines. Proof of human–mouse coexistence dates back to the beginning of agriculture societies. Ancient documentation on regulatory food standards specifying the maximum amount of rodent excreta in wheat suggests that the first contacts with MMTV could date back thousands of years and might have occurred through the oral route. Even nowadays, according to FDA guidelines, processed foods are allowed to contain low levels of rodent-related material [118]. Additional observations supporting this thesis are the reports of cases of MMTV-exposed laboratory workers with immune reactivity to the virus [124] and a case of seroconversion with the development of an invasive ductal BC [125]. Interestingly, Stewart et al. also observed that the incidence of human BC was highest in areas where *Mus domesticus* was the resident species of house mouse [126,127]. This evidence suggests that a high frequency of *Mus domesticus* could be related to increased incidence of HMTV, thus supporting that the virus could be introduced into the human population by mice. The Authors correlated the high copy numbers of endogenous *Mtv* in murine species with the potential of generating exogenous, infectious virions by recombination between different *Mtvs*, as clearly demonstrated in an experimental setting by Golovkina et al. [128,129]. In 2004, Levine et al. [130] came to the same conclusion as Stewart et al. [126,127] by showing that there was a significantly similar geographic prevalence of MMTVels in humans and MMTV in wild mice. A recent study by Wang et al., in China, found similar evidence, showing that the difference in the prevalence of MMTV-like virus in BC samples from northern (23%) and southern China (6%) could be dependent on the distribution of *M. domesticus*, *M. musculus*, and *M. castaneus* [131]. From this perspective, the spread of house mice could parallel the geographical variation in the prevalence of MMTVels-positive human BC. In contrast to this hypothesis, Lessi et al. [14] came to a different conclusion: given the evidence that (a) human milk does not appear as a convincing source of infection [111,132]; (b) MMTVels has been identified in human saliva and salivary glands [133]; and (c) there is no convincing data supporting a mouse-to-man viral transmission, they concluded that exogenous HMTV would be the result of cross-species transmission from mouse to man, followed by inter-human spread. Again, further studies are needed to clarify this aspect.

## 8. HMTV in Human BC

Over time, several studies have attempted to investigate the clinical, molecular, and pathological parameters of MMTV-like positive breast tumors and searched for a correlation between the presence of the virus and human proteins already known to be associated with BC, such as progesterone receptors (PgR), estrogen receptors (Er), c-erbB-2, and p53. Already in the 1990s, Pogo et al. [134] tried to investigate the association of MMTV-positive BC with c-erbB-2, p53, bcl-2, progesterone receptors, and cathepsin D and had tried to classify MMTV-positive tumors within one of the two subsets of BC known at that time [135]. They found that 90% of positive tumors were invasive ductal carcinomas [136] and that MMTVels presence did not correlate with the expression of any protein. Moreover, these tumors were not significantly associated with any one subset. Of note, they did find a positive correlation with the expression of laminin receptor, a marker for invasiveness and poor prognosis [134]. 

In 2004, observations by Lawson et al. and Faedo et al. [137,138] of elevated p53 expression in a number of MMTVels-positive BC samples supported the idea that HMTV may be associated with breast carcinogenesis. This hypothesis was also reinforced by the evidence that p53 is associated with aggressive carcinogenesis [137]. Moreover, Faedo et al. succeeded in demonstrating an association between the presence of MMTV-like viral sequences and the expression of progesterone receptors, while no significant association was found with the expression of estrogen receptors [137,138]. These observations are not surprising, because hormone-response elements are present in MMTV LTRs amplified from human BC [139,140]. The idea that HMTV is associated with more aggressive kinds of tumors appeared to be more and more consistent: Ford et al. [141] reported a high prevalence of MMTV-like sequences in more invasive BC tissues from Australian women. Levine et al. suggested a possible association between the presence of the MMTVels and tumor aggressiveness by using a tumor-aggressiveness classification system. In particular, a high prevalence of MMTVels was found in inflammatory BC from Tunisian patients [130]. Starting from these observations, Wang et al. [142] focused their attention on gestational BC, finding a large proportion of these cancers associated with MMTV-like sequences detections. BC discovered during pregnancy or lactation is known to be very aggressive and of poor prognosis, probably because of the effect of hormones on the stimulation of cellular growth [143,144]. However, in contrast with the previous reports, although triple-negative BC (TNBC) is considered one of the most aggressive ones among the molecular subtypes of BC [145,146], Gupta et al. [147] unexpectedly found that MMTVels were present in only 7% of TNBC samples from a Croatian cohort, and similar findings (4%) came from a Brazilian TNBC cohort [148].

Due to the close parallel between the biology of mouse mammary tumors and MMTV-like associated human BC, Lawson et al. [149] hypothesized that the histological patterns should be similar between the two species. Indeed, they succeeded in demonstrating that there was significant correlation between the presence of MMTV in human BC and histological characteristics resembling those of MMTV-positive murine mammary tumors, which were mostly classified as Dunn type B and Dunn type A, thus confirming the observations of Nartey et al. [150]. At the time, no correlation was found between the presence of viral sequences, the histological grade, and a number of associated biomarkers, such as ER, PR, c-erbB-2, and p53 by the same authors, confirming previous results by Pogo et al. [134]. Recently, however, Wang et al. succeeded in demonstrating a significant association between the prevalence of HMTV sequences and c-erbB-2 expression [131].

## 9. MMTV in Other Human Tumors and Diseases

It is known that MMTV may also induce tumors other than mammary cancer, predominantly hematological lymphocyte malignancies, in mice [74,151]. Similarly, MMTVels have been detected in human lymphomas [152,153,154] and in hepatocellular carcinomas and various other liver diseases [155]. Earlier studies had identified MMTV-like antigens and sequences in human milk [156], saliva [133], and peripheral blood lymphoid cells [157]. This indicates that, as in mice, MMTV-like sequences may also be found in organs other than the breast in humans. 

Viral sequences from a human betaretrovirus (known as HBRV) sharing 91–99% nucleotide identity with MMTV have been associated with primary biliary cirrhosis (PBC). The first evidence of this dates back to 1998, when Mason et al. found anti-retroviral antibodies in patients with PBC and other idiopathic biliary disorders [158]. In 2004, the same group succeeded in cloning a retroviral sequence from PBC lymph nodes and identified a virus very closely resembling MMTV [159]. Johal et al. [157] extended these observations: starting from the evidence that HMTV-associated BC was correlated with the expression of a number of carcinogenic cellular factors [137,138], they investigated a putative association between hepatic diseases showing MMTV-like sequences and expression of PgR, estrogen receptor-α (ER-α), and nuclear accumulation of the p53 protein, already known to be implicated in hepatic carcinogenesis [160,161]. These scientists succeeded in demonstrating that MMTV-like sequences could be associated with PBC in Australian patients, with a prevalence of 12%, while viral sequences were never found in normal tissue, supporting the already existing literature suggesting that, in this case, HBRV was exogenous rather than endogenous. Moreover, no correlation between the expression of ER-α, PgR, and nuclear expression of p53 was found. A more recent study by Mason et al. [162] suggested that HBRV could reside in the biliary epithelium, reporting frequent detection of HBRV RNA (from 59% to 75%, depending on the method of investigation) and proviral integrations (58%) in biliary epithelial cells from patients with PBC. In contrast, viral integrations were found in only 13% of PBC liver samples, confirming that HBRV integrations were rarely found in whole liver samples. Moreover, they also found an increased frequency of HBRV integrations and evidence of HBRV RNA in patients with autoimmune hepatitis and cryptogenic liver disease [162].

MMTVels were also detected in human lymphoma. Etkind et al. reported data from a number of cases of patients diagnosed with both BC and lymphoma, positive for MMTV-like virus sequences [153,154]. While MMTV is mainly associated with T-cell lymphoma in murine species [72,163,164], they found positive lymphomas with both B and T phenotype. They suggested that the expansion of human B and T lymphocytes, occurring after the stimulation of TCR by viral particles, provided so many cells for infection that they could not only allow the virus to be transmitted to mammary cells, but also to act as an etiological agent for the lymphoid tissue itself. This added to the already existing evidence that, in PBC-affected patients, the viral particles were preferentially localized in lymphoid tissue, while less particles were found in the liver, the target organ for the disease [165]. Further proof suggesting that mammary cancer and lymphoma could share a common etiology came from the evidence of an increased incidence of murine lymphomas when mice were either inoculated with human mammary cancer extracts [166] or implanted with human inflammatory BC cells [167]. Interestingly, it has also been shown that women with prior lymphoma diagnosis have double the risk of BC compared to women without previous lymphoma [152]. However, Etkind et al. [153] found MMTVels in tumoral samples from patients diagnosed with lymphoma alone. Finally, beside PBC and lymphoma, MMTV-like sequences were detected in various other human cancers, including endometrial carcinoma [168], liver, ovary, prostate, and skin [107], but the role of MMTV in these cancers was not clearly defined.

## 10. MMTV-Like Virus in Companion Animals

BC clearly has a multifactorial etiology. The attempt to correlate the incidence of human BC with socioeconomic and geographic parameters has yielded contradictory conclusions [126,127,169,170,171]. Because the prevalence of MMTV-like sequences in humans varies between different ethnic groups [8,105,139,172,173,174], the model of an infectious exogenous agent combined with immunogenetic factors in the etiology of BC would fit best [175]. Several environmental risk factors have been proposed for human sporadic BC, in addition to the presence of MMTV-like sequences. Strikingly, as already discussed, high incidence of human BC has been found to correlate with the predominant presence of a particular mouse strain [126,127]. Based on this correlation, transmission of MMTV from this mouse strain to man was proposed [126,127]. However, if we accept that mice can transmit MMTV to humans, there appears to be a missing link that may be represented by pets such as dogs and cats [10,11,12,13,15,176]. In this hypothesis, humans would acquire or have acquired an MMTV-like virus through contact with their infected pets. The first evidence that MMTV could infect feline cells came in 1976 [177]. In the same year, Vaidya et al. [154] confirmed that MMTV was able to productively infect feline kidney cells. Next, Howard and Schlom [178] demonstrated that MMTV-like viral particles isolated from feline cells, following serial passages, have the ability to productively infect cultured cells derived from different hosts, including feline, canine, bat, mink, murine, and human cells. It has also been shown that the MMTV virus is able to broaden its host range through recombination of sequences of the exogenous virus and endogenous betaretroviruses, or between different strains of endogenous or exogenous viruses. Later, these findings were confirmed by Golovkina et al. [128,129]. However, other feline retroviruses have also been reported to be able to infect human cells [179,180], possibly increasing the likelihood of promoter/enhancer insertions associated with retrovirus-mediated tumorigenesis. To date, though, this ability has not been shown to be necessarily correlated with causing disease [24].

The current literature reports very few cases of MMTV-like sequence amplification in samples from dogs [12], whereas most of the literature regarding investigations on pets reported the amplification of MMTV-like sequences from feline lymphoid tissue or mammary carcinoma [11,12,13,15]. Human exposure to cats may be just as ancient as it is to mice. Indeed, while agricultural societies flourished, cats were domesticated from wild felines exactly with the goal to protect grains from rodents. Nowadays, in countries of higher socio-economic standards, having cats as pets remains a frequent habit. On the other hand, in countries of lower socio-economic standards, contact with stray cats, which are likely to feed on mice regularly, is more frequent. Cats might acquire MMTV from mice, primarily by feeding on MMTV-infected mice. Just like any exogenous infection, MMTV virions may be taken up from preys and then transmitted to offspring through milk, or while grooming and smearing fur with saliva that could contain the virus. A significant number of families, often with children, have cats as pets, and contact with these pets can lead to infection for humans: humans may come into direct contact with infectious secretions, such as saliva from pets smearing fur with saliva or licking their owners’ hands. Infectious secretions on human hands would provide a ready route for oral transmission. Alternatively, humans may also be inadvertently scratched or bitten by their pets [10]. Studying feline saliva could clarify if it could really be a compatible vehicle for infection, as suggested for human saliva by Mazzanti et al. [133].

## 11. MMTV-Like Virus Associated with Feline and Canine Mammary Tumors

In 2005, Szabo et al. found, for the first time, viral sequences highly similar to MMTV (>90%) in tissue from cats [11]. Notably, these were found in the thymus of a kitten, while weak amplification was reported from the spleen of an adult of unknown age. No amplification was ever reported from feline non-lymphoid organs, such as the kidney, or from dog spleens. These observations led Szabo et al. [11] to the idea that the MMTV-like sequence-positive kitten was comparable to a suckling mouse, being at the end of the first phase of infection by MMTV, when the virus spreads from Peyer’s patches to other organs, or in the second phase of infection, when all lymphoid organs are involved in a Sag-dependent immune response. These Authors were also the first to report a marked difference in the level of amplification and in the ease or difficulty of viral isolation between samples from different species, normally complicated by a high background in amplification reactions visualized in agarose gels under UV light. The results obtained by Szabo et al. [10] brought to light the hypothesis that cats might be the intermediate species between mouse and man in the transmission of the virus. Further evidence came from Hsu et al. [12], who succeeded in detecting MMTV-like sequences in samples coming from feline and canine tissues. Already in 2006, Laumbacher et al. [175] had speculated that there was a correlation between women who developed BC and women who owned pets. They found out that women with BC owned dogs, but not cats, significantly more often as compared to age-matched controls, thus suggesting that dogs may also be involved in the spread of the virus. Hsu et al. showed the presence of MMTVels and LTR sequences in canine and feline malignant mammary tumors in Chinese pets, with an incidence of 3.49% for *env* sequences (3/86) and 18.60% (16/86) for LTR in dogs and 22.22% (2/9) for both *env* and LTR sequences in cats.

Through phylogenetic analysis, they ruled out amplification of endogenous viruses and showed that MMTV *env* sequences from mice, dogs, cats, and humans were in separate clusters, indicating that the viruses differ between these hosts. Sequences from dogs and cats showed a similarity ranging from 94% to 98%, when compared to those from mice and humans. Nevertheless, MMTV-like *env* and LTR sequences were also detected in canine benign mammary tumors, in tissue from healthy dogs, and in normal tissue from cats and dogs bearing mammary tumors. Therefore, the presence of MMTV-like sequences was not significantly correlated to the pathological and immunohistochemical characteristics of tumors, such as clinical stage; histological type and grade of mammary tumor; and expression of progesterone, estrogens, and c-erbB-2 receptors. Despite this, they also succeeded in detecting env transcript in one of the MMTV-positive canine samples, adding an important support to the hypothesis of the exogenous spreading of the virus by pets. In contrast with the results of the Chinese study, an Italian study by Civita et al. [13] succeeded in detecting MMTVels only in feline mammary tumors, with no amplification from canine mammary tumors or from the tissue of healthy dogs and cats. Differently from most tissues used in previous medical and veterinary studies, these samples were formalin-fixed and paraffin-embedded (FFPE); therefore, the investigation was carried out using fluorescent PCR and subsequent fragment analysis, a method that has proven to be sensitive and robust in detecting MMTV-like sequences and was specially designed to avoid issues in FFPE tissues [181]. MMTVels were found in 7% (6/86) of cats examined and showed a homology of 97% and 99% for HMTV and MMTV, respectively. Particularly, one sample showed a polymorphism leading to a previously unreported amino-acid substitution (Thr > Ala). Phylogenetic analysis testified that MMTVels isolated from Italian cats did not belong to *Felis catus* endogenous retroviruses (FcERV) or the HERV-K family. The sequences detected in mice, humans, and dogs were also classified into clusters different from MMTV, suggesting that MMTV can be transmitted between these hosts. As already reported by the Chinese study, Civita et al. did not detect a significant correspondence between the presence of viral sequences and the histological type and grade of tumors [14]. To confirm their data, they highlighted the presence of MMTV-like p14 protein, the signal peptide of MMTV envelope precursor localized in the nucleolus of infected cells [119], exclusively in MMTV-like-positive sequences samples. In a subsequent study, Parisi et al. [15] found three MMTV-like virus-positive samples from FFPE-retrieved feline mammary carcinomas from another area, with a prevalence of 12,5%. The new sequences showed a similarity of 98% and 100% to HMTV and MMTV, respectively. Again, phylogenetic analysis confirmed the results of the previously mentioned research. The expression pattern of p14 was analyzed as in the previous study (Figure 2).

Thorough analyses of the molecular portrait of MMTV-like positive mammary carcinomas was carried out by Parisi et al. in an attempt to establish a possible correlation between the presence of the viral sequences and tumor parameters, but, to date, no correlation was found between the presence of viral sequences in tumors and their histotype, histological grading, or molecular phenotype of PCR-positive feline mammary carcinomas.

Starting from the evidence that (i) MMTV is responsible not only for mammary carcinoma, but also for lymphoma in susceptible mice, (ii) MMTV-like sequences were found in human lymphoma [153,154], leading to the hypothesis of a common etiology between BC and lymphoma [152], and (iii) MMTV-like sequences were amplified from the thymus of a kitten and the spleen of a cat (even if isolated only from the former) [11], Parisi et al. turned their attention to feline lymphomas [manuscript submitted]. Amplification of MMTVels in five out of 53 FIV/FELV-negative FFPE feline lymphomas (with a prevalence of 9.4%) was detected. Feline lymphomas showing amplification in the *env* region were classified as T cell gastrointestinal lymphoma (*n* = 2), B-cell gastrointestinal lymphoma (*n* = 1), and B-cell nasal lymphomas (*n* = 2). It is remarkable that the anatomical sites from which MMTVels were amplified communicate directly with the environment; in particular, the gastroenteric system is the first entry and infection site for the virus in mice. In addition, as highlighted by Velin et al. [182], the nasal lymphoid tissue of adult mice could also act as an entry site for productive infection by MMTV. Expression of p14, the signal peptide, was exclusively detected in neoplastic cells from PCR-positive samples, mostly in the cytoplasm and rarely in the nuclei (Figure 3).

Supplementary anamnesis was collected by interviewing pet owners and referring the clinicians of cats with MMTVels-positive lymphomas. Data collected highlighted that all the subjects had have, in some way, a history of contact with the external environment, and, in two cases, cats sharing the environment with PCR-positive subjects had died, in turn, because of lymphoma.

## 12. Diagnosis, Prevention, and Therapy: New Perspectives on the Role of p14

The potential for the MMTV-like virus as a cause of human BC has opened a new era for prevention and therapy. There are still very few studies in this field, but recent literature has focused on the role of p14. P14, a multifunctional 98-amino-acid peptide, is the signal peptide of the MMTV envelope precursor, initially named MMTV-p14 (p14, for short) because of its electrophoretic mobility. Its function is to mediate the translocation of the 74-kDa envelope protein precursor of MMTV across the membrane of the endoplasmic reticulum (ER). Following entry in the ER, the signal sequence is cleaved, and the envelope protein is further glycosylated and processed to give viral gp52 and gp36 envelope glycoproteins. Usually, after fulfilling their ER-targeting function, signal peptides are degraded by peptidase. However, as in this case, p14 signal peptide might have additional functions [183] because it was found to be associated with the nucleoli of cells from murine mammary carcinoma and lymphoma that harbored the virus [184,185,186] and in human BC [187], with nucleo-cytoplasmic shuttling [119]. It was suggested that p14 affects the tumorigenic potential of MMTV-infected cells in its phosphorylated state. The envelope precursor signal peptide would function as a tumor-modulating phosphoprotein, phosphorylated in cells by two serine kinases, CK2 at serine 65, and PKC at serine 18. Phosphorylation of serine 65 is associated with enhanced tumorigenicity, while phosphorylation of serine 18 is associated with decreased tumorigenicity [119]. In conclusion, the oncogenic potential of MMTV would be based on the CK2 to PKC ratio. Furthermore, it was demonstrated that p14 is also the signal sequence of a splice variant of MMTV-env (termed REM) with a key role as a nuclear export factor for intron-containing transcripts (known as REM, an analogous of the HIV-Rev protein) [24,25], thus classifying MMTV as a complex virus.

P14 is also expressed on the surface of MMTV-associated murine and human cells [188] and may have a unique therapeutic potential due to its immunogenicity, suggesting a number of potential strategies for the targeting and treatment of MMTV-associated cancers. In 2016, Braitbard et al. [188] showed that p14 could be successfully employed in active immunization, using p14 or mutant forms of it, or in passive immunization using anti-p14 monoclonal antibodies. Moreover, cytotoxic T lymphocytes specific for p14-cell surface epitopes isolated from MMTV-associated tumors were suggested to be potentially useful for successful therapy. Intracellular introduction of immune conjugates (p14 monoclonal antibodies and a translocator protein) into MMTV-associated cancer cells or intracellular introduction of small molecule inhibitors of p14 have also been suggested for treatment purposes. Finally, p14 and anti-p14 antibodies could also be used in assays for early diagnosis of MMTV-associated tumors. Even if p14 has recently been detected immunohistochemically in more and more MMTVels-positive cases, additional studies are needed to shed further insight into these innovative possibilities.

## 13. Conclusions and Perspectives

Because the incidence of human BC is increasing and there is present compelling evidence that HMTV may be involved in its tumorigenesis, the effort to investigate HMTV-positive neoplasms and the virally related immunological alterations in further depth should go forward in order to develop more effective prevention, control, and therapeutic strategies. Moreover, because a zoonotic hypothesis has been proposed, the presence of MMTVels and p14-positive neoplasms in cats, a species with a close social interaction with humans, raises concerns on the possible involvement of this species in the epidemiology of MMTV. Further investigations are needed to solidify these observations.

## Figures and Tables

**Figure 1 viruses-14-00977-f001:**
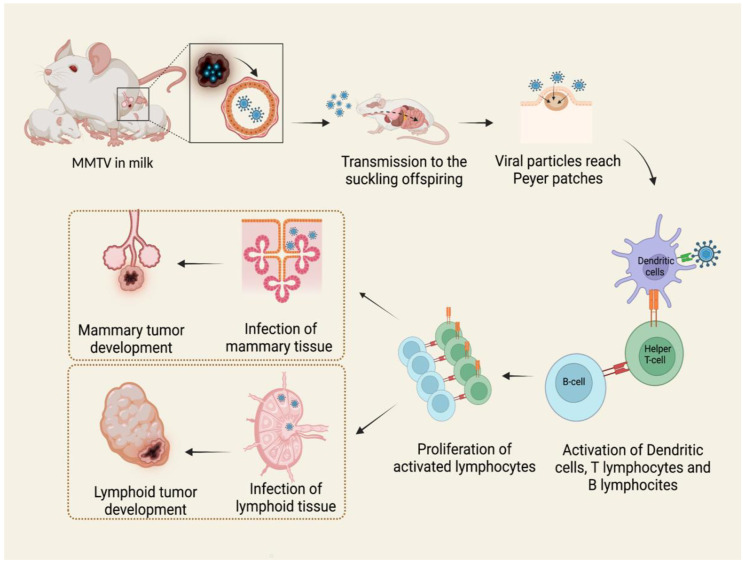
MMTV exogenous cycle. Viral particles transmitted through milk from mother to offspring infect antigen presenting cells in Peyer’s patches of the gut. The subsequent activation of B lymphocytes and T helper lymphocytes causes the expansion of different immune cell clones and the proliferation of both infection-competent and infected cells. MMTV-infected lymphocytes carry the virus to mammary glands and other tissues, among which are lymphoid tissue, causing neoplasms by different mechanisms. Created by Biorender.com accessed on 14 March 2022.

**Figure 2 viruses-14-00977-f002:**
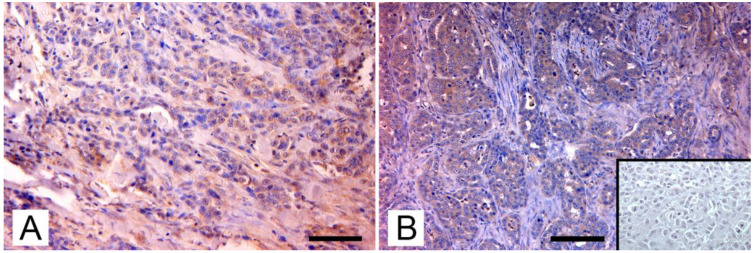
Immunohistochemical expression of p14 MMTV *env* protein in feline mammary carcinomas: (**A**) Mammary carcinoma. Weak cytoplasmatic immunostaining in neoplastic epithelial cells (arrows; IHC, bar = 50 μm); (**B**) Mammary carcinoma. Diffuse intense cytoplasmic staining of neoplastic cells (arrows), which form tubular structures and solid growth patterns. Several atypical mitoses are detectable. Insert: negative control, absence of cytoplasmic staining in neoplastic mammary cells. (IHC, bar = 50 μm). The IHC was performed by incubating slides with 1:2000-diluted rabbit polyclonal antibody against MMTV-p14 (kindly provided by Dr J. Hochman, University of Jerusalem, Israel) for 2 h at room temperature and following the Labeled Streptavidin Biotin method.

**Figure 3 viruses-14-00977-f003:**
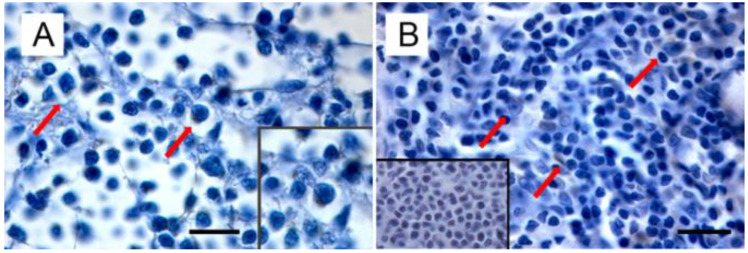
Immunohistochemical expression of p14 in MMTVels-positive feline lymphoma: (**A**) nasal B-cell lymphoma. Diffuse cytoplasmic expression of p14 in tumoral lymphoblasts (arrows). Neoplastic cells show severe anisokaryosis, anisocytosis, and reticulate chromatin. Insert: high magnification of P14-positive lymphoblast. (IHC, bar = 100 μm); (**B**) gastrointestinal T-cell lymphoma. Scattered cytoplasmic expression of p14 in neoplastic lymphoblasts (arrows). Tumoral cells show marked anisokaryiosis and anisocytosis, indented nuclei with marginated chromatin, and prominent nucleoli. Insert: negative control. Absence of cytoplasmic staining in neoplastic lymphoblast. (IHC, bar = 50 μm). The IHC was performed by incubating slides with 1:2000-diluted rabbit polyclonal antibody against MMTV-p14 (kindly provided by Dr J. Hochman, University of Jerusalem, Israel) for 2 h at room temperature and following the Labeled Streptavidin Biotin method.

## Data Availability

Not applicable.

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
