# Peer review of "Mouse Mammary Tumor Virus (MMTV) and MMTV-like Viruses: An In-depth Look at a Controversial Issue"

_viruses, 2022, doi:10.3390/v14050977_

Round 1
Reviewer 1 Report
The review by Parisi et al. on the contribution of MMTV-like betaretrovirus to human breast cancer, lymphoma and other human diseases is quite thorough. The emphasis of this review is to highlight the possibility that the exogenous mouse virus might be acquired from pets, particularly cats and dogs. Overall, the review synthesizes recent results and makes valid points. However, some of the language needs to be toned down. Below, I detail changes and suggestions that might improve the manuscript.
In the Abstract, line 20, “the hypothesis of an MMTV-like raised” has a word missing. Perhaps “virus” between “MMTV-like” and “raised”. Also, “an MMTV-like” would read better.
Section 1, line 50, please delete “species” after human. Also, please cite the study by Luc Montagnier’s group (Crépin et al. 1984, Biochem Biophys Res Commun;118(1):324-31).
Please right justify Section 2.
In Figure 1, for the 3rd step, it would be preferable to keep the present tense (viral particles reach Peyer’s patches).
Line 63, the authors mention hormone responsive elements in the MMTV LTR, but MMTV LTRs also contain several regulatory sequences that bind to specific transcription factors involved in the control of cell proliferation (PMID: 12207913). This should be mentioned, as mechanisms relevant to mammary tumorigenesis and lymphomagenesis likely involve TEAD (formerly called TEF-1) transcription factors.
Section 4
Line 174, sentence is cumbersome. I would delete “Differently”. Start with “Viral integration events occur infrequently within gene coding regions, often at a significant distance from the target gene, up to 200 kb away”
Line 193, “found upstream or downstream of Rorc and Myc” Also, please italicize all mouse genes, per the convention.
Section 11, line 584, has several problems. Here, the authors are no longer talking about cats, but human lymphoma “subjects” (should be plural) who provided medical accounts (anamnesia) of having been exposed to cats (not clear whether the cats died of lymphoma, also). Lines 584-587 should either be deleted or reworded to make clear this refers to human subjects.
Figure 2. Please provide details about the P14 antibody (source, cat. Number) and the method of detection.
Section 12, line 589: the words “demonstration of” should be deleted and perhaps replaced by “potential for”. There is still not solid proof that MMTV-like virus causes breast cancer in humans. It can only be proven by reduction in breast cancer incidence following global vaccination, as proven for human papillomavirus.
Similarly, section 13 should be toned down. Despite being in the “pro” camp, this reviewer takes exception with the statement that “it is well established that HMTV is involved in its (human BC) tumorigenesis”. Perhaps “there is compelling evidence” would be a more defensible position.
Author Response
Reviewer 1
The review by Parisi et al. on the contribution of MMTV-like betaretrovirus to human breast cancer, lymphoma and other human diseases is quite thorough. The emphasis of this review is to highlight the possibility that the exogenous mouse virus might be acquired from pets, particularly cats and dogs. Overall, the review synthesizes recent results and makes valid points. However, some of the language needs to be toned down. Below, I detail changes and suggestions that might improve the manuscript.
In the Abstract, line 20, “the hypothesis of an MMTV-like raised” has a word missing. Perhaps “virus” between “MMTV-like” and “raised”. Also, “an MMTV-like” would read better.
Text has been modified accordingly
Section 1, line 50, please delete “species” after human. Also, please cite the study by Luc Montagnier’s group (Crépin et al. 1984, Biochem Biophys Res Commun;118(1):324-31).
Text has been modified accordingly. Reference has been added.
Please right justify Section 2.
Section 2 has been justified.
In Figure 1, for the 3rd step, it would be preferable to keep the present tense (viral particles reach Peyer’s patches).
The tense has been modified in figure 1.
Line 63, the authors mention hormone responsive elements in the MMTV LTR, but MMTV LTRs also contain several regulatory sequences that bind to specific transcription factors involved in the control of cell proliferation (PMID: 12207913). This should be mentioned, as mechanisms relevant to mammary tumorigenesis and lymphomagenesis likely involve TEAD (formerly called TEF-1) transcription factors.
A paragraph about TEAD transcription factors has been added.
Section 4
Line 174, sentence is cumbersome. I would delete “Differently”. Start with “Viral integration events occur infrequently within gene coding regions, often at a significant distance from the target gene, up to 200 kb away”
Text has been modified accordingly
Line 193, “found upstream or downstream of Rorc and Myc” Also, please italicize all mouse genes, per the convention.
Text has been modified properly
Section 11, line 584, has several problems. Here, the authors are no longer talking about cats, but human lymphoma “subjects” (should be plural) who provided medical accounts (anamnesia) of having been exposed to cats (not clear whether the cats died of lymphoma, also). Lines 584-587 should either be deleted or reworded to make clear this refers to human subjects.
In that paragraph we was referring to cats as well, so the sentences have been properly reformulated to make the concept more clear.
Figure 2. Please provide details about the P14 antibody (source, cat. Number) and the method of detection.
Data required have been added.
Section 12, line 589: the words “demonstration of” should be deleted and perhaps replaced by “potential for”. There is still not solid proof that MMTV-like virus causes breast cancer in humans. It can only be proven by reduction in breast cancer incidence following global vaccination, as proven for human papillomavirus.
Text has been modified accordingly
Similarly, section 13 should be toned down. Despite being in the “pro” camp, this reviewer takes exception with the statement that “it is well established that HMTV is involved in its (human BC) tumorigenesis”. Perhaps “there is compelling evidence” would be a more defensible position.
Text has been modified accordingly
Reviewer 2 Report
This study reviews the controversial area of the potential zoonosis of MMTV into the human species and its role in human breast cancer. It is an extensive review with 169 references that covers a broad area of MMTV biology and its transmission into humans. The first part of the review covers a brief history of MMTV, its classification, genome, transcripts, and life cycle (both exogenous and endogenous), role of mammary stem cells in tumorigenesis, mechanism of mammary and lymphoid tumors, and role of host factors and viral genes in susceptibility and resistance to infection. The second part covers studies related to MMTV transmission into humans. This section covers MMTV in both human breast cancer and other tumors as well as other human diseases like primary biliary cirrhosis. Furthermore, it also discusses evidence of the presence of MMTV into other animals like cats and dogs that are human pets as possible missing link in the zoonosis of MMTV into humans. Finally, it discusses the role of the signal peptide of Env (p14) as a possible tumorigenic protein of MMTV whose presence in infected cells could be exploited to target MMTV-associated cancers for therapy.
As can be seen, this is quite an extensive review covering aspects of MMTV and HMTV studies that so far have not been brought together in one review in this manner. However, given the existing controversial nature of the existence of HMTV, this reviewer finds that some assertions of the authors need to be toned down where statements are made as if the existence of HMTV is no longer debatable and is a proven fact. Furthermore, the article needs to be reviewed by a native English speaker to make corrections related to language.
Major Points:
Title: Better to modify the title to: “MMTV and MMTV-like Viruses: an in-depth Look at a Controversial Issue”
Lines 22-23: Modify sentence as follows: Finally, the possibility of a MMTV-like virus as a cause of human BC could open a new era ….
Line 68: MMTV transcripts code for Gag, Pro-dUTPase, and Pol. The function of dUTPase (dUT) in viral replication is not clear and may have to do with its replication in non-dividing cells.
Line 70: Env gene encodes three protein chains, not two: signal peptide p14, SU gp52, and TM gp36.
Line 87: Clarify what is meant by “infection-competent” cells and how they differ from “infected” cells.
Line 101: What is meant by “sub-viral form”. MMTV is cell associated and intracytoplasmic particles can be detected.
Line 109: Double check “lung” tissue. It normally does not express MMTV similar to heart.
Line 111: Elaborate on “with a few exceptions”. What are those exceptions?
Line 120-121: It is better to state that “MMTV exists as two forms in mice” rather than saying that “MMTV has two routes of transmission in vivo”. Change exogenous “route” to “virus”.
Lines 142-146: Delete these lines since they are repeated earlier (lines 97-100).
Line 589: This has not be “demonstrated” yet. Change as follows: “The suggestion that an MMTV-like virus could be a cause of human BC has opened a ….
Lines 627-628: Delete the clause: “it is well established that HMTV is involved in its tumorigenesis”.
Minor Points:
- Line 12: Add “a” after as.
- Lines 13-16. Divide the long sentence in two. Replace “and” in line 14 with “that” and put a period after divide. Delete “while” and start next sentence with “Simultaneously, it avoids immune responses and causes tumors ….
- Line 24-25. The abstract can be reworded better by moving the last line of the abstract before “Furthermore” on line 20.
- The names of viral and cellular genes should be written properly (italicized or capitalized) throughout the manuscript according to the species.
- Line 79. Cite Figure 1 after “epithelium”.
- Line 327: Add latest reference of Stewart and Chen published in Viruses where they have revisited this hypothesis. Also replace “high concentration” with “the range of”.
- Line 339: Replace “diffusion” by “spread”.
- Line 360. Add ref. 113 at the end.
- Line 361: I believe reference 114 should be 117 and 118.
- Line 364: Is reference 115 correct? Double check.
- Line 424: Replace “supplied” with “provided”.
- Line 431: Delete “was observed”.
- Line 441: Delete “data on the”.
- Line 449: Add ref. 106 and the newer Stewart and Chen reference after Mus domesticus.
- Line 469: Change “in feline” to “from feline”.
- Line 487: Replace “to feline” with “with feline”.
- Line 528: Replace “so” with “therefore”.
- Line 531: Do you mean reference 160?
- Lines 545 and 567:5% and 12.4%. Which % is correct?
- Line 550. Replace “Cat” with “Feline” mammary gland.
- Line 553:
- Line 556: Replace “in the” with “in an”.
- Lines 566-570: Is it “five” or “eight”? Pls clarify. Move the numbers to end of the classification as follows: gastrointestinal lymphoma (n =3) T-cell lymphomas (n = 2), B-cell lymphoma (n = 1), and B-cell nasal lymphomas (n =2).
- Line 567: FIV/FELV
- Line 570: Change “district” to “sites or regions”.
- Line 578. Delete “Cat”. MMTVels-positive “feline” lymphoma.
- Line 617: … anti-p14 ….
- Line 632: Replace “so” with “a”.
- Line 634: Replace “… go in depth on this virus” with “solidify these observations”.
Author Response
Reviewer 2
This study reviews the controversial area of the potential zoonosis of MMTV into the human species and its role in human breast cancer. It is an extensive review with 169 references that covers a broad area of MMTV biology and its transmission into humans. The first part of the review covers a brief history of MMTV, its classification, genome, transcripts, and life cycle (both exogenous and endogenous), role of mammary stem cells in tumorigenesis, mechanism of mammary and lymphoid tumors, and role of host factors and viral genes in susceptibility and resistance to infection. The second part covers studies related to MMTV transmission into humans. This section covers MMTV in both human breast cancer and other tumors as well as other human diseases like primary biliary cirrhosis. Furthermore, it also discusses evidence of the presence of MMTV into other animals like cats and dogs that are human pets as possible missing link in the zoonosis of MMTV into humans. Finally, it discusses the role of the signal peptide of Env (p14) as a possible tumorigenic protein of MMTV whose presence in infected cells could be exploited to target MMTV-associated cancers for therapy.
As can be seen, this is quite an extensive review covering aspects of MMTV and HMTV studies that so far have not been brought together in one review in this manner. However, given the existing controversial nature of the existence of HMTV, this reviewer finds that some assertions of the authors need to be toned down where statements are made as if the existence of HMTV is no longer debatable and is a proven fact. Furthermore, the article needs to be reviewed by a native English speaker to make corrections related to language.
Major Points:
Title: Better to modify the title to: “MMTV and MMTV-like Viruses: an in-depth Look at a Controversial Issue”
The title has been changed as suggested
Lines 22-23: Modify sentence as follows: Finally, the possibility of a MMTV-like virus as a cause of human BC could open a new era ....
The sentence has been modified accordingly
Line 68: MMTV transcripts code for Gag, Pro-dUTPase, and Pol. The function of dUTPase (dUT) in viral replication is not clear and may have to do with its replication in non-dividing cells.
The sentence has been modified.
Line 70: Env gene encodes three protein chains, not two: signal peptide p14, SU gp52, and TM gp36.
Text has been modified accordingly
Line 87: Clarify what is meant by “infection-competent” cells and how they differ from “infected”
cells.
Infected cells are those activated from the contact through the viral antigen directly (like APCs), infection competent cells are those activated by the antigen associated with class II MHC of APCs (like CD4+ T lymphocites). This terminology has been previously reported by Ross, 2010.
Line 101: What is meant by “sub-viral form”. MMTV is cell associated and intracytoplasmic particles can be detected.
Text has been modified to clarify.
Line 109: Double check “lung” tissue. It normally does not express MMTV similar to heart.
MMTV infection in lung, together with references 34-37, have been reported from Callahan and smith, 2000.
Line 111: Elaborate on “with a few exceptions”. What are those exceptions?
Text has been modified since “few exceptions” was referred to another part of the sentence.
Line 120-121: It is better to state that “MMTV exists as two forms in mice” rather than saying that “MMTV has two routes of transmission in vivo”. Change exogenous “route” to “virus”.
Text has been modified accordingly
Lines 142-146: Delete these lines since they are repeated earlier (lines 97-100).
Text has been modified accordingly
Line 589: This has not be “demonstrated” yet. Change as follows: “The suggestion that an MMTV- like virus could be a cause of human BC has opened a ....
Text has been modified accordingly
Lines 627-628: Delete the clause: “it is well established that HMTV is involved in its tumorigenesis”.
We would like to maintain some reference about HMTV in the first sentence to better introduce the conclusion. We modified the sentence as suggested from reviewer 1.
Minor Points:
- Line 12: Add “a” after as.
Text has been modified accordingly
- Lines 13-16. Divide the long sentence in two. Replace “and” in line 14 with “that” and put a
period after divide. Delete “while” and start next sentence with “Simultaneously, it avoids
immune responses and causes tumors ....
Text has been modified accordingly
- Line 24-25. The abstract can be reworded better by moving the last line of the abstract before “Furthermore” on line 20.
Text has been modified accordingly
- The names of viral and cellular genes should be written properly (italicized or capitalized)
throughout the manuscript according to the species.
The names of viral and cellular genes have been written properly
- Line 79. Cite Figure 1 after “epithelium”.
Text has been modified accordingly
- Line 327: Add latest reference of Stewart and Chen published in Viruses where they have
revisited this hypothesis. Also replace “high concentration” with “the range of”.
The reference has been added and the text has been changed as suggested
- Line 339: Replace “diffusion” by “spread”.
Text has been modified accordingly
- Line 360. Add ref. 113 at the end.
Ref 113 has been added
- Line 361: I believe reference 114 should be 117 and 118.
References have been corrected
- Line 364: Is reference 115 correct? Double check.
Reference 115 has been removed from that paragraph
- Line 424: Replace “supplied” with “provided”.
Text has been modified accordingly
- Line 431: Delete “was observed”.
Text has been modified accordingly
- Line 441: Delete “data on the”.
Text has been modified accordingly
- Line 449: Add ref. 106 and the newer Stewart and Chen reference after Mus domesticus.
References have been added
- Line 469: Change “in feline” to “from feline”.
Text has been modified accordingly
- Line 487: Replace “to feline” with “with feline”.
Text has been modified accordingly
- Line 528: Replace “so” with “therefore”.
Text has been modified accordingly
- Line 531: Do you mean reference 160?
Yes, it was a typos error.
- Lines 545 and 567: 12.5% and 12.4%. Which % is correct?
There was a mistake, since the second percentage was about lymphomas. It has been corrected.
- Line 550. Replace “Cat” with “Feline” mammary gland.
Text has been modified accordingly
- Line 553: Patterns.
Text has been modified accordingly
- Line 556: Replace “in the” with “in an”.
Text has been modified accordingly
- Lines 566-570: Is it “five” or “eight”? Pls clarify.
It is five. The other three comes from a previous study, the missing reference was added.
Move the numbers to end of the classification as follows: gastrointestinal lymphoma (n =3) T-cell lymphomas (n = 2), B-cell lymphoma (n = 1), and B-cell nasal lymphomas (n =2).
Text has been modified accordingly
- Line 567: FIV/FELV
Text has been modified accordingly
- Line 570: Change “district” to “sites or regions”.
Text has been modified accordingly
- Line 578. Delete “Cat”. MMTVels-positive “feline” lymphoma.
Text has been modified accordingly
- Line 617: ... anti-p14 ....
Text has been modified accordingly
- Line 632: Replace “so” with “a”.
Text has been modified accordingly
- Line 634: Replace “... go in depth on this virus” with “solidify these observations”.
Text has been modified accordingly
Reviewer 3 Report
This is a relatively comprehensive review of the potential involvement of HMTV in human tumors, in particular in breast cancer. It could be improved by including more discussion of the mechanism by which HMTV might cause breast cancer since (unlike in mice) to date there is no compelling data that integration of the retroviral DNA causes clonal amplification of human breast cells. Indeed, there are a number of other mechanisms (some of them indirect) by which other retroviruses are involved in causing malignancies (see below). I think if this can be included in the paper then it should be acceptable for publication. There are a few other issues as follows:-
- The title might be better as “Mouse mammary tumor virus (MMTV) and MMTV-Like Viruses: an Indepth Look Inside a Controversial Issue”
- Page 2, line 66: Indik et al., 2005, Virology 337, 1-6 and Mertz et al., 2005, Journal of Virology, 79, 14737-14747 would be better references for MMTV being a complex retrovirus
- Page 3, line 97: “The long period of latency between the ingestion of infected milk and mammary tumor development suggests that MMTV does not contain an oncogene” is okay but the long latency does not per se suggest that MMTV “causes the dysregulation of host gene expression through the integration of its provirus into the host genome”. This part of the sentence should be deleted, also since it is explained in more detail later.
- Page 5, line 176: In the sentence “Even if further studies are needed, several pieces of evidence suggested that MMTV-induced mammary tumors were the result of different multiple integrations” should be reworded to something like ““Even if further studies are needed, several studies have suggested that MMTV-induced mammary tumors are the result of different multiple integrations”
- Page 7, line 309: The statement “Nowadays, these uncertainties have been definitively overcome thanks to the studies by Lessi and colleagues…” is an overstatement and needs to be weakened to something like “The recent studies by Lessi and colleagues have demonstrated that a MMTV-like betaretrovirus has been present in humans since the Copper Age. These studies took specific precautions to rule out the possibility of contamination and of false-positives due to endogenous-retrovirus integration into the human genome”.
- Page 7, line 320: “Ancient documentation on regulatory food standards specifying the maximum amount of rodent excreta in wheat suggest that the first contacts with MMTV could date back thousands of years and might have occurred through the oral route.” This sentence could be strengthened by adding “Even nowadays, according to FDA guidelines, processed foods are allowed to contain low levels of rodent-related material (Salmons and Gunzburg, 2013, Revisiting a role for a mammary tumor retrovirus in human breast cancer. Int J. Cancer 133, 1530–1536).
- Given that next generation sequencing failed to find MMTV sequences in breast cancer tumors (see Tang K-W, Larsson E. 2017 Tumour virology in the era of high-throughput genomics. Phil. Trans. R. Soc. B 372: 20160265. http://dx.doi.org/10.1098/rstb.2016.0265 and also http://larssonlab.org/tcga-viruses/report_BRCA.php), it seems unlikely that (in contrast to mice) integration of MMTV is driving the genesis of breast cancer. Nevertheless, as discussed by Salmons and Gunzburg, 2015, (Tumorigenesis Mechanisms of a Putative Human Breast Cancer Retrovirus. Austin Virol and Retrovirology. 2015; 2(1): 1010. https://austinpublishinggroup.com/virology/fulltext/avrv-v2-id1010.php) mechanisms other than insertion for tumorigenicity have been documented for other retroviruses. These include
- MMTV may express other, as yet unknown, proteins that may be involved in tumorigenicity, perhaps even encoded by the minus strand – a precedent here is the HBZ product of HTLV-1.
- Already identified proteins may have additional functions in analogy to the envelope proteins of other viruses. Accessory factors such as Rem, Sag, p14, Naf, or Env may also potentially play a role, as might the immunoreceptor tyrosine-based activation (ITAM) motif in the envelope gene originally described by Katz et al. as transforming mammary epithelial cells grown in three-dimensional culture and in mice and an infectious MMTV provirus lacking a functional Env ITAM motif developed mammary tumors with decreased incidence and higher latency (Dudley J.P., Golovkina, T.V.; Ross, S.R. 2016, Lessons Learned from Mouse Mammary Tumor Virus in Animal Models. ILAR J 662, 57, 12-23). More recent data suggests a more subtle effect resulting from the ITAM domain suppressing apoptosis through ITAM-mediated Src tyrosine kinase signaling (Salmons B and Gunzburg WH, 2013, Revisiting a role for a mammary tumor retrovirus in human breast cancer. Int J. Cancer 133, 1530–1536).
- MMTV infection of cells might transiently activate latent human breast cancer associated virus such as EBV and HPV, possibly in a similar manner to the way that Human Immunodeficiency Virus (HIV) indirectly activates KSHV in Kaposi’s sarcoma.
- Interaction between MMTV/HMTV and human endogenous retroviruses may also play a role in the causation of breast cancer.
General points
- The authors might want to include some sentences about the influence of the microbiome on the gut mediate transfer of MMTV (Wilks J., Golovkina T. (2012).Influence of microbiota on viral infections. PLoS Pathog. 8:e1002681 and Gunzburg W.H. and Salmons B. With a little help from my enteric microbial friends. Front. Microbiol. 2015, 6: 1029. doi: 3389/fmicb.2015.01029)
- some of the terminology used is inconsistent e.g. rem or REM for the name of the gene
- the English grammar needs checking
Author Response
Reviewer 3
This is a relatively comprehensive review of the potential involvement of HMTV in human tumors, in particular in breast cancer. It could be improved by including more discussion of the mechanism by which HMTV might cause breast cancer since (unlike in mice) to date there is no compelling data that integration of the retroviral DNA causes clonal amplification of human breast cells. Indeed, there are a number of other mechanisms (some of them indirect) by which other retroviruses are involved in causing malignancies (see below). I think if this can be included in the paper then it should be acceptable for publication. There are a few other issues as follows:-
- The title might be better as “Mouse mammary tumor virus (MMTV) and MMTV-Like Viruses: an Indepth Look Inside a Controversial Issue”
- The title has been modified accordingly
- Page 2, line 66: Indik et al., 2005, Virology 337, 1-6 and Mertz et al., 2005, Journal of Virology, 79, 14737-14747 would be better references for MMTV being a complex retrovirus
- References have been added
- Page 3, line 97: “The long period of latency between the ingestion of infected milk and mammary tumor development suggests that MMTV does not contain an oncogene” is okay but the long latency does not per se suggest that MMTV “causes the dysregulation of host gene expression through the integration of its provirus into the host genome”. This part of the sentence should be deleted, also since it is explained in more detail later.
- Text has been modified accordingly
- Page 5, line 176: In the sentence “Even if further studies are needed, several pieces of evidence suggested that MMTV-induced mammary tumors were the result of different multiple integrations” should be reworded to something like ““Even if further studies are needed, several studies have suggested that MMTV-induced mammary tumors are the result of different multiple integrations
- Text has been modified accordingly
- Page 7, line 309: The statement “Nowadays, these uncertainties have been definitively overcome thanks to the studies by Lessi and colleagues…” is an overstatement and needs to be weakened to something like “The recent studies by Lessi and colleagues have demonstrated that a MMTV-like betaretrovirus has been present in humans since the Copper Age. These studies took specific precautions to rule out the possibility of contamination and of false-positives due to endogenous-retrovirus integration into the human genome”.
- Text has been modified accordingly
- Page 7, line 320: “Ancient documentation on regulatory food standards specifying the maximum amount of rodent excreta in wheat suggest that the first contacts with MMTV could date back thousands of years and might have occurred through the oral route.” This sentence could be strengthened by adding “Even nowadays, according to FDA guidelines, processed foods are allowed to contain low levels of rodent-related material (Salmons and Gunzburg, 2013, Revisiting a role for a mammary tumor retrovirus in human breast cancer. Int J. Cancer 133, 1530–1536).
- Text has been modified accordingly, reference has been added.
- Given that next generation sequencing failed to find MMTV sequences in breast cancer tumors (see Tang K-W, Larsson E. 2017 Tumour virology in the era of high-throughput genomics. Phil. Trans. R. Soc. B 372: 20160265. http://dx.doi.org/10.1098/rstb.2016.0265 and also http://larssonlab.org/tcga-viruses/report_BRCA.php), it seems unlikely that (in contrast to mice) integration of MMTV is driving the genesis of breast cancer. Nevertheless, as discussed by Salmons and Gunzburg, 2015, (Tumorigenesis Mechanisms of a Putative Human Breast Cancer Retrovirus. Austin Virol and Retrovirology. 2015; 2(1): 1010. https://austinpublishinggroup.com/virology/fulltext/avrv-v2-id1010.php) mechanisms other than insertion for tumorigenicity have been documented for other retroviruses. These include
- MMTV may express other, as yet unknown, proteins that may be involved in tumorigenicity, perhaps even encoded by the minus strand – a precedent here is the HBZ product of HTLV-1.
- Already identified proteins may have additional functions in analogy to the envelope proteins of other viruses. Accessory factors such as Rem, Sag, p14, Naf, or Env may also potentially play a role, as might the immunoreceptor tyrosine-based activation (ITAM) motif in the envelope gene originally described by Katz et al. as transforming mammary epithelial cells grown in three-dimensional culture and in mice and an infectious MMTV provirus lacking a functional Env ITAM motif developed mammary tumors with decreased incidence and higher latency (Dudley J.P., Golovkina, T.V.; Ross, S.R. 2016, Lessons Learned from Mouse Mammary Tumor Virus in Animal Models. ILAR J 662, 57, 12-23). More recent data suggests a more subtle effect resulting from the ITAM domain suppressing apoptosis through ITAM-mediated Src tyrosine kinase signaling (Salmons B and Gunzburg WH, 2013, Revisiting a role for a mammary tumor retrovirus in human breast cancer. Int J. Cancer 133, 1530–1536).
- MMTV infection of cells might transiently activate latent human breast cancer associated virus such as EBV and HPV, possibly in a similar manner to the way that Human Immunodeficiency Virus (HIV) indirectly activates KSHV in Kaposi’s sarcoma.
- Interaction between MMTV/HMTV and human endogenous retroviruses may also play a role in the causation of breast cancer.
Several further data have been reported on mechanisms other than insertion for tumorigenicity
General points
- The authors might want to include some sentences about the influence of the microbiome on the gut mediate transfer of MMTV (Wilks J., Golovkina T. (2012).Influence of microbiota on viral infections. PLoS Pathog. 8:e1002681 and Gunzburg W.H. and Salmons B. With a little help from my enteric microbial friends. Front. Microbiol. 2015, 6: 1029. doi: 3389/fmicb.2015.01029)
A paragraph has been added.
- some of the terminology used is inconsistent e.g. rem or REM for the name of the gene
We have checked the terminology throughout the manuscript.
- the English grammar needs checking
English grammar has been checked
Round 2
Reviewer 1 Report
The authors have responded to this reviewer’s concerns adequately. However, there are many typographical and grammatical errors throughout the text that warrant correction.
The following list should be addressed:
Fist, there is an inappropriate use of hyphenation throughout. For example, line 2, please do not hyphenate viruses. I suggest removing the hyphenation function altogether.
Line 51, humans
Line 63-65, these sentences read poorly, and a space is needed after the period. “enhancers” should be plural and not hyphenated. Ditto, elements on line 65.
Line 68, tissues
Line 74, “CDP”
Line 90, please remove hyphenation
Line 107, add comma after “In blood”
Line 239, “not”
Line 262, “TLR7 and(or the adaptor/s thereof” doesn’t make sense. Should it be “TLR7 and/or the adaptors thereof”?
Section 7 could use some airing out. Could a “hard” return start a new paragraph at line 281? And also at line 297?
Line 327, why introduce a new format for references here? “[Callahan et al., 2012] [67]” Also, there needs to be a space after [67]. I would simply use the number in brackets, unless particular emphasis on a group’s findings warrants mentioning.
Line 336, Cheng should be capitalized, and reference starts with a bracket and ends with a parenthesis. Again, should this be emphasized, or could we simply use the number in brackets?
Line 339, can a new paragraph start here?
Line 374, insert space after period.
Line 497, immunogenetic is a preferred spelling
Line 559, remove the hyphen between MMTV-els (to be consistent with the rest)
Line 637, insert space after period.
Line 683, insert space after “and”
Author Response
The authors have responded to this reviewer’s concerns adequately. However, there are many typographical and grammatical errors throughout the text that warrant correction.
The following list should be addressed:
First, there is an inappropriate use of hyphenation throughout. For example, line 2, please do not hyphenate viruses. I suggest removing the hyphenation function altogether.
The hyphenation function has been removed.
Line 51, humans
The text has been revised accordingly.
Line 63-65, these sentences read poorly, and a space is needed after the period. “enhancers” should be plural and not hyphenated. Ditto, elements on line 65.
The text has been modified accordingly.
Line 68, tissues
The text has been corrected.
Line 74, “CDP”
The text has been corrected.
Line 90, please remove hyphenation
The text has been corrected.
Line 107, add comma after “In blood”
The text has been corrected.
Line 239, “not”
The text has been corrected.
Line 262, “TLR7 and(or the adaptor/s thereof” doesn’t make sense. Should it be “TLR7 and/or the adaptors thereof”?
The text has been clarified.
Section 7 could use some airing out. Could a “hard” return start a new paragraph at line 281? And also at line 297?
Two paragraphs have been inserted.
Line 327, why introduce a new format for references here? “[Callahan et al., 2012] [67]” Also, there needs to be a space after [67]. I would simply use the number in brackets, unless particular emphasis on a group’s findings warrants mentioning.
The names of the authors in brackets had been added only to check that the numbers entered have been correct, therefore they are removed throughout the text.
Line 336, Cheng should be capitalized, and reference starts with a bracket and ends with a parenthesis. Again, should this be emphasized, or could we simply use the number in brackets?
The names of the authors in brackets had been added only to check that the numbers entered have been correct, therefore they are removed throughout the text.
Line 339, can a new paragraph start here?
A new paragraph has been inserted.
Line 374, insert space after period
A space has been inserted.
Line 497, immunogenetic is a preferred spelling
The text has been modified accordingly.
Line 559, remove the hyphen between MMTV-els (to be consistent with the rest)
The hyphen has been removed.
Line 637, insert space after period.
A space has been inserted.
Line 683, insert space after “and”
A space has been inserted.